# Machine Learning-Based Dynamic Modeling for Process Engineering Applications: A Guideline for Simulation and Prediction from Perceptron to Deep Learning

Carine M. Rebello [1,2], Paulo H. Marrocos [1], Erbet A. Costa [2], Vinicius V. Santana [1], Alírio E. Rodrigues [1], Ana M. Ribeiro [1] and Idelfonso B. R. Nogueira [1,*]

1 Laboratory of Separation and Reaction Engineering, Associate Laboratory LSRE/LCM, Department of Chemical Engineering, Faculty of Engineering, University of Porto, Rua Dr. Roberto Frias, 4200-465 Porto, Portugal; carine.menezes@ufba.br (C.M.R.); phmarrocos@gmail.com (P.H.M.); up201700649@edu.fe.up.pt (V.V.S.); arodrig@fe.up.pt (A.E.R.); apeixoto@fe.up.pt (A.M.R.)
2 Departamento de Engenharia Química, Escola Politécnica (Polytechnic School), Universidade Federal da Bahia, Salvador 40210-630, Brazil; erbetcosta@ufba.br
* Correspondence: idelfonso@fe.up.pt

**Abstract:** A misusage of machine learning (ML) strategies is usually observed in the process systems engineering literature. This issue is even more evident when dynamic identification is performed. The root of this problem is the gradient explode and vanishing issue related to the recurrent neural networks training. However, after the advent of deep learning, these issues were mitigated. Furthermore, the problem of data structuration is often overlooked during the machine learning model identification in this field. In this scenario, this work proposes a guideline for identifying ML models for the different applications in process systems engineering, which are usually for simulation or prediction purposes. While using the proposed guideline, the work also identifies a virtual analyzer for a pressure swing adsorption unit. In these types of adsorption separations, it is usual that the measurement of the main properties is not done online. Therefore, the virtual analyzer is another contribution of this manuscript. The overall results demonstrate that even though the test provides good performance during the ML model identification, its quality might degenerate over the application domain if the model application is overlooked.

**Keywords:** machine learning; deep learning; dynamic modeling; pressure swing adsorption

## 1. Introduction

While, from a first perspective, the concepts of prediction and simulation appear to be the same, they are different, and this difference must be emphasized. A simulation is performed to verify the response of a model using input data and initial conditions. The values calculated as a response of the model have a sampling time equal to the sampling time of the input variables. A prediction computes the response of a model at some specified future horizon of time through the projection of current and past values of measured input and output values, as well as initial conditions. Thus, a simulation does not require measurements (the actual states of the system) beyond the initial condition. In contrast, a prediction depends on it, leading simulations to be adequate to situations in which measurements are unavailable, such as control system design, fault detection, and process optimization [1]. On the other hand, in scenarios where there is a measurable quantity, but its measurement has a significant deadtime, prediction is more suitable.

The difference between these concepts reflects the instruments and techniques adequate for one or the other. The data structure is the first and essential step in identifying the models when dealing with data-driven modeling. The predictors are structures used to organize the data following its application. There are many predictor structures [2].

However, the most important to represent nonlinear systems are the nonlinear autoregressive with exogenous inputs (NARX) and the nonlinear output error (NOE) [3]. The first depends on past measurements of the input and the output, the "exogenous inputs" fraction of its name. It assumes that the error associated with its prediction comes from the information provided by the measurements used as input of the structure. The NOE does not depend on past measures of the output to make forecasts. It uses its past forecasts and the measurement of the inputs; additionally, it assumes that the error is only added to its output. It forms the actual output value.

Consequently, the NARX nature of evaluating the error makes it inadequate for long-term simulations. To perform predictions, it must be assumed that the forecast made at one time is the measurement input of a future time; the model, in this manner, is used recursively. Hence, the error of one prediction is carried to another time as the error is associated with a measurement input, and, consequently, the error has a cumulative nature. In contrast, the internal structure of the NOE is noiseless, and the error is only added to the prediction: as such, it does not have a cumulative nature, and it is, therefore, more adequate to simulations [4].

Despite this inadequacy, the NARX structure is frequently used in chemical and process engineering to simulate dynamic systems, mainly due to its simplicity of identification. The NOE structure is less studied due to an increased difficulty associated with its identification [3,4].

Suppose an artificial intelligence is utilized as nonlinear function to implement the NARX or NOE structure. In that case, the NARX model is easier to train as it is possible to use it in a series-parallel architecture. It allows the utilization of the feedforward training strategy of the static backpropagation [5]. The NOE model does not use the output measurement to compute predictions. Instead, it uses its past predictions; it can only be trained in a parallel architecture. This is a complex issue, as the parameters of the models depend on the model output, which becomes an input in the next iteration. Thus, it is vulnerable to exploding or vanishing gradients, requiring more sophisticated neural network structures to prevent this [6]. The advent of deep learning is intrinsically related to this issue.

The regular inappropriate use of the NARX model to perform simulations in the chemical engineering domain has led to the lack of investigation of strategies to determine the best choice of a recurrent model approach. While the NARX can be utilized reasonably in certain cases, its use must be evaluated, as the cumulative error associated with its structure can lead to unrealistic results or instability of the model [1].

In this scenario, this work addresses this open issue in the literature providing guidelines for the adequate use of machine learning models. As a study case, the empirical modeling of a pressure swing adsorption (PSA) unit is presented.

The PSA can be depicted as a separation process that takes advantage of the interaction between different chemical species in a fluid phase and a solid adsorbent to separate these species. This interaction's intensity is leveraged by the variation of the system's pressure to achieve the separation [7,8].

Machine learning techniques have been applied in pressure swing adsorption units to address the problems found in this field. For instance, Ye et al. (2019) [9] used a feedforward structure to develop a simulation model for a PSA unit. Tong et al. (2021) [10] presented an artificial neural network model to optimize a PSA unit. In Sant Anna et al. (2017) [11] and Subraveti et al. (2019) [12], they also applied a feedforward structure to model a PSA unit for optimization purposes. These works have important contributions to the PSA field, addressing the process optimization and modeling issues. However, the authors of the previously referred works applied a predictors structure without evaluating the predictor embedding dimensions or without considering the fact that the used predictor is not appropriate to a simulation scenario. In a broader context of chemical engineering, it is possible to find the same tendency. For instance, Meleiro et al. (2009) [13] propose modeling a chemical plant through ANN tools. Mouellef et al. (2021) [14] presents a chromatography process design and operation optimization aided by artificial intelligence. Rahnama et al.

(2020) [15] applied ML tools to model a basic oxygen steelmaking from experimental data. Even in other engineering fields, ML strategies are usually employed in this sense. Coccia et al. (2021) [16] employed ANN models to simulate the cooling demand of a single-family house. Pervez et al. (2021) proposes an ANN model to predict wind speed to be applied as a simulation model in a control scheme. Still, these works do not evaluate the suitable predictor, embedding dimensions, and ML model for their applications. As it is possible to see from this literature review, there is overall misleading employment of machine learning models in the field of chemical engineering, and more specifically, in the area of modeling pressure swing adsorption unit. In fact, it is generally challenging to find works in chemical engineering that apply machine learning models and evaluate the issues related to the predictors. As pointed by Dobbelaere et al. (2021) [17]: "The greatest threat in artificial intelligence research today is inappropriate use because most chemical engineers have had limited training in computer science and data analysis.". Therefore, this work addresses part of this issue, providing a comprehensive guideline for ML application in this field.

The dynamics of a PSA unit are very complex, where no steady state is observed. Furthermore, it presents difficulties in obtaining a measurement of its main properties, such as concentrations. These measurements are usually obtained using offline instruments as gas chromatography (GC) techniques. These instruments require a significant amount of time to perform the measurement, during which the desired information is unknown.

In this scenario, this work proposes identifying a soft sensor to perform real-time predictions of the purities and recoveries of a PSA unit. On the other hand, it presents the identification of machine learning (ML) models to simulate the dynamic behavior PSA unit. The machine learning strategies employed here were the traditional feedforward neural network (FNN), a recurrent neural network (RNN), and a deep neural network (DNN) [18]. Therefore, three main contributions of this work are highlighted. A general contribution is the discussion of guidelines for the dynamic identification of ML models in chemical and process engineering. Overall, two specific contributions are the development of a soft sensor for a PSA unit and a simulator of the PSA process.

## 2. Methods

This section presents the main methods applied in developing the ideas addressed here.

### 2.1. Study Case

Cyclic adsorption processes promote the separation of complex mixtures while being an energetically efficient and environmentally friendly route. Among the cyclic adsorption processes, the pressure swing adsorption (PSA) unit can be highlighted for its wide range of applications to promote gas-phase separations. These processes leverage the variation of the adsorption capacity of a given adsorbent with the pressure, hence promoting the gas separation by taking advantage of the adsorption phenomena. The unit operates in a series of discrete steps, each with a specific function, as described below. This creates a dynamic behavior where no steady estate is reached.

The PSA unit of this work will be used as a pre-treatment of syngas to purify it. As shown in Figure 1, the PSA unit has five steps: steps I and IV are conducted under varying pressure, whereas the others are performed at constant pressure. It will be fed a gas mixture of $CO_2$, CO, and $H_2$ to capture the $CO_2$ and separate it from the CO and $H_2$, considering the components of the purified syngas. This operation aims to achieve a $CO_2$-enriched stream with an $H_2$/CO stoichiometric ratio between 2.2 and 2.3, the suitable feed range for the Fischer–Tropsch step.

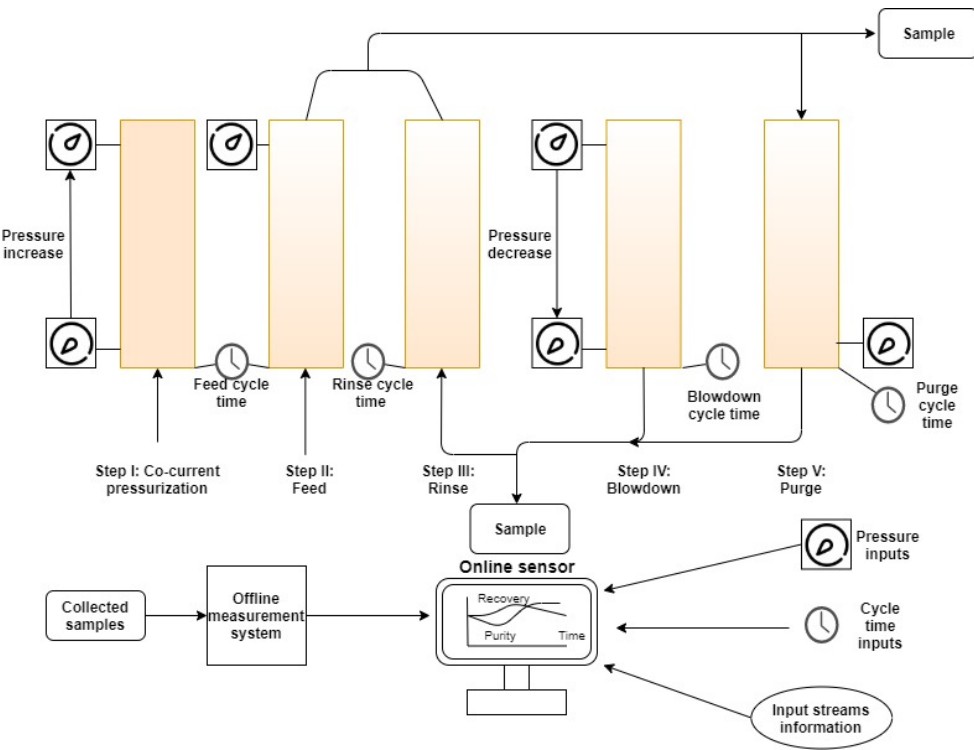

**Figure 1.** Representation of the proposed measurement system for the PSA unit.

A virtual plant of this unit was used based on its phenomenological model. The virtual plant will generate the synthetic data with which the networks will be trained. The adequate tools are discussed in the next section. The phenomenological model used in this work was proposed by Silva et al. (1999) [19] and validated by Regufe et al. [20], in which the following assumptions were made:

1. Gas-phase follows an ideal behavior;
2. The flow is axially dispersed;
3. There are external mass and heat transfer resistances, represented by the film model;
4. There is an internal mass transfer resistance, represented by the linear driving force (LDF) model;
5. The heat transfer in the solid phase is faster than in the gas phase to the extent that there are no temperature gradients inside the particles;
6. The porosity along the bed is constant;
7. The Ergun equation is valid locally.

### 2.2. Simulation Case: Short-Term Simulation for Online Applications

Simulations are usually necessary for optimization and control ends. For instance, the literature on PSA optimization and control presents works addressing the optimization and control of these units through simulations performed using ML models [21,22]. Furthermore, generally in the chemical engineering literature, it is challenging to find works that properly use ML models for simulation ends. Usually, NARX structures are employed for this end. However, no works were found that address the proper employment of ML models to perform this task. Therefore, it is a pertinent issue to be evaluated. It provides essential guidelines to the literature on simulation using ML models.

The virtual plant will serve as a benchmark simulation to compare the simulations performed in this case. As the phenomenological model requires great computational effort to simulate the unit, a surrogate model might be more appropriate when simulations are necessary to be performed in the short term.

### 2.3. Prediction Case: Online Sensor

The online sensor proposed in this work will provide information about the system in real-time to help with the problem of measurement dead time associated with a PSA unit. The virtual plant will simulate the offline measurements provided to the network in the prediction case. It will be used to predict the following values of the properties until the subsequent measurement is performed again to provide information during the dead time. Figure 1 illustrates the online sensor: samples of the output will be regularly obtained to measure the concentrations and get the purity and recovery of the PSA unit. Additionally, the input information of the PSA unit, such as cycle times, column pressures, and flow values, will also be made available to the online sensor.

It is expected that the error associated with each prediction of the online sensor rises as time passes. The machine learning models used to represent the nonlinear dynamic system of the PSA will be based on deep neural networks, recurrent neural networks, and feedforward neural networks. Therefore, these three main ML strategies will be evaluated in this prediction scenario, hence, a comprehensive comparison of these models will be conducted.

### 2.4. Predictors

A dynamic system can be represented as a time series, a series of data points ordered in time. This series can be described in a general form by Equation (1):

$$y(t) = G[u(t), u(t-1), \ldots, u(t-k), v(t), v(t-1), \ldots, v(t-k)] \tag{1}$$

in which $u$ is an input, and $v$ is the white noise, and $y$ is an output. There are many tools to the development of models, one of which is the predictors. Though many predictors exist [23], only a few of them are the ones to which a detailed explanation will be given hereinafter. The relationship between past inputs $u$ and the output $y$ can be given by Equation (2):

$$y(t) = g(\phi(t), \theta) + v(t) \tag{2}$$

in which $g$ is a nonlinear function, for example, a neural network; $\phi(t)$ is the regression vector, and $\theta$ is the function's parameters. The choice of different components of the regression vector determines the predictor structure.

### 2.5. Nonlinear Autoregressive with Exogenous Inputs (NARX)

The components of the regression vector associated with the NARX structure are the past measured inputs and the past measured outputs. As such, through this structure, a prediction $\hat{y}$ of the actual output, $y$ is given by Equation (3).

$$\hat{y}(t) = g[y(t-1), \ldots, y(t-n_a), u(t-d), \ldots, u(t-d-n_b+1), \theta] + v(t) \tag{3}$$

in which $d$ is the input delay; and $n_a$ and $n_b$ are the number of past values for the output and input, respectively. Note that the noise is added inside the structure in itself: for the NARX, it is assumed that the error information is filtered through the system's dynamic [1]. Consequently, the error structure of NARX is additive. Equation (3) is a one-step-ahead prediction. However, it often is extended to multiple step-ahead predictions or simulations by using the predictions $\hat{y}$ instead of measurements $y$ as input to make future predictions recursively.

Thus, the error associated with a prediction adds to future predictions, as it is carried over through the additive nature of the error in this structure: it is here that the problem lies. Figure 2 shows the NARX structure used recursively. The prediction is used instead of measurement as input of the function. Note that the number of measured information, output predictions, and the input time delay are parameters to be considered when developing the structure: in Figure 2, there is only one measured input, one output prediction, and an input time delay of one, to simplify the example.

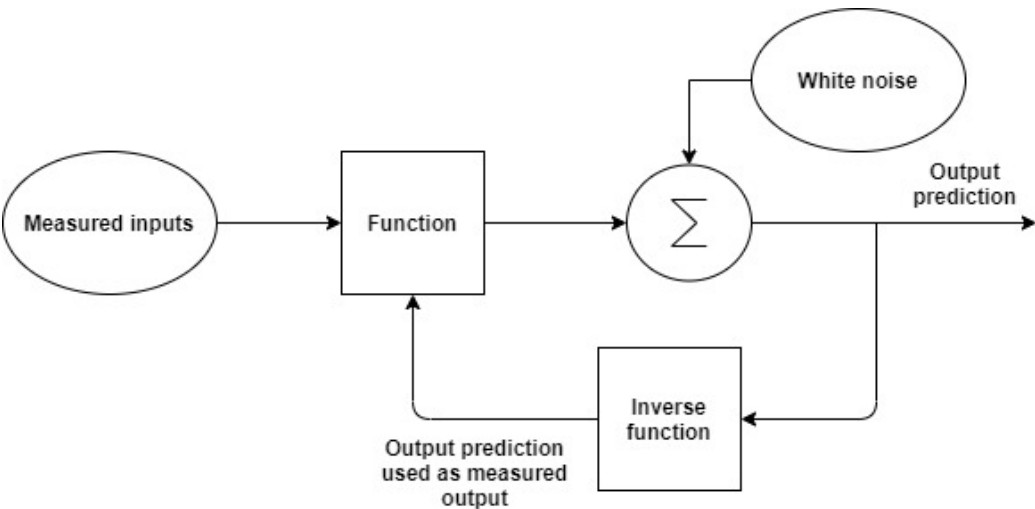

**Figure 2.** NARX structure used recursively.

In other words, if a simulation is performed with NARX, the structure will be used recursively. Thus, the predictions calculated at one time are used as measurements at the next instant. This means that there is an assumption of knowing the system's state at future times. In the absence of the measurement, the past prediction is assumed to be a good representation of the predictions made. This assumption might not be reasonable in simulations, as there is no previous knowledge of future states.

*2.6. Nonlinear Output Error (NOE)*

The NOE structure does not require an exogenous input as a component of the regression vector: it uses the measured input and the predicted output. With the objective of more easily organizing the equations to be shown, for the NOE, the predicted output will be represented by $p(t)$. As such, Equations (4) and (5) represent a one-step-ahead prediction through NOE.

$$p(t) = g[p(t-1), \ldots, p(t-n_a), u(t-d), \ldots, u(t-d-n_b+1)] \tag{4}$$

$$y(t) = p(t) + v(t) \tag{5}$$

As can be seen, the prediction structure is noiseless, and the noise is added only to the prediction after the system. Consequently, if this structure is used recursively to perform long-term predictions or simulations, the error is not propagated through each prediction. The easiest way to visualize an NOE structure is by analogy with a system of partial differential equations that represents a rigorous model. As such, the NOE is more adequate than the NARX to perform simulations. Figure 3 exhibits the NOE structure used recursively: it can be seen that the noise is only added after the prediction steps to compose the actual output value. Similar to the NARX structure, the number of measured inputs, output predictions, and the input delay are parameters to be determined. In order to simplify the description shown in the figure, only output prediction and one recursive delay were chosen.

Thus, the NARX structure has the additive noise issue and, as such, must be treated carefully when used for long-term predictions or simulations. However, this care is not usual in the literature. Hence, the necessity to deep discuss this issue.

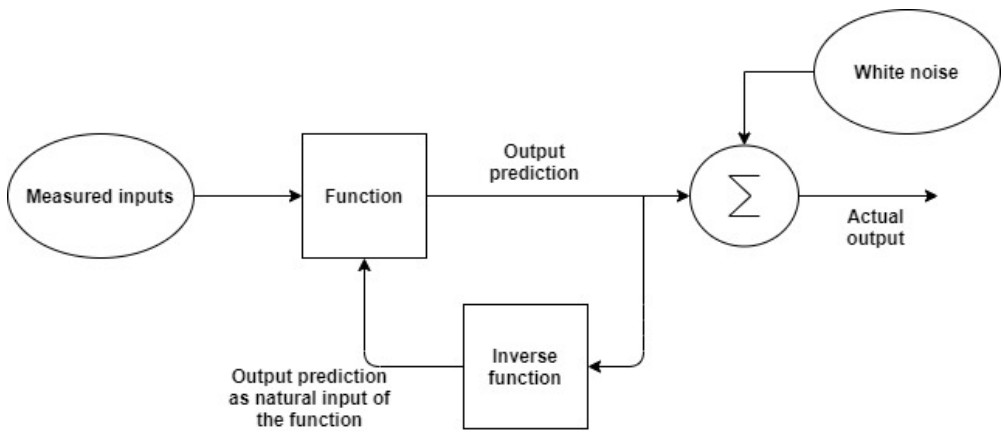

**Figure 3.** NOE structure used recursively.

Although inadequate for such means, the NARX is often used to simulate as a consequence of a more accessible training process, as previously described. Furthermore, the NOE approach for dynamic ML modeling was impossible until the last decade due to software/hardware limitations. This strategy became available to model time-series only after the advent of deep learning. Therefore, it is still usual the systematic use of NARX strategies coped with ML modeling. While this approach can sometimes lead to good results, it can also create a significant difference between the predicted and correct values due to the addictive nature of the error [4].

## 3. Analysis, Results, and Discussion

This section provides the analysis, the correspondent result, and the two applications cases that illustrate the topics discussed in this work.

### 3.1. Data Acquisition

The quality of the predictions made by the network depends heavily on the quality of the data used to train it, as it is the basis over which the model learns the phenomena in analysis. As such, to produce the data set, the Latin hypercube sampling (LHS), a method developed by McKay et al. [24], was used. In LHS, a sample range of each input variable was divided into a given number of intervals with equal probability, and a random value inside each interval was chosen. The input variables were randomly organized to form input groups to calculate the output values. This method, which is an extension of stratified sampling, ensures that the input variables have all their portions represented and all portions of their distributions. In this article, the LHS designed 1000 experiments. Each experiment was applied to the virtual plant. The results for each experiment were recorded until the PSA reached the cyclic steady state, after 25 cycles. Therefore, 25,000 points were generated.

All input variables were considered in the design of experiments, namely: inlet temperature, purge flowrate, rinse flow rate, high pressure, low pressure, feed step duration, rinse step duration, and purge step duration. The limits used was based on Nogueira et al. (2020) [25], and are respectively: $\theta_{min}$ = [304 K, 0.225 SLPM, 0.425 SLPM, 3.4 bar, 0.55 bar, 380 s, 187 s, 80 s] and $\theta_{max}$ = [350 K, 0.345 SLPM, 0.575 SLPM, 5 bar, 1.1 bar, 680 s, 253 s, 110 s].

The training, validation, and test data set was created using the PSA virtual plant. The input data generated by the LHS is presented in Figure 4. In the diagonal of this figure, it is possible to observe that the samples are uniformly distributed within the sampling space, as expected.

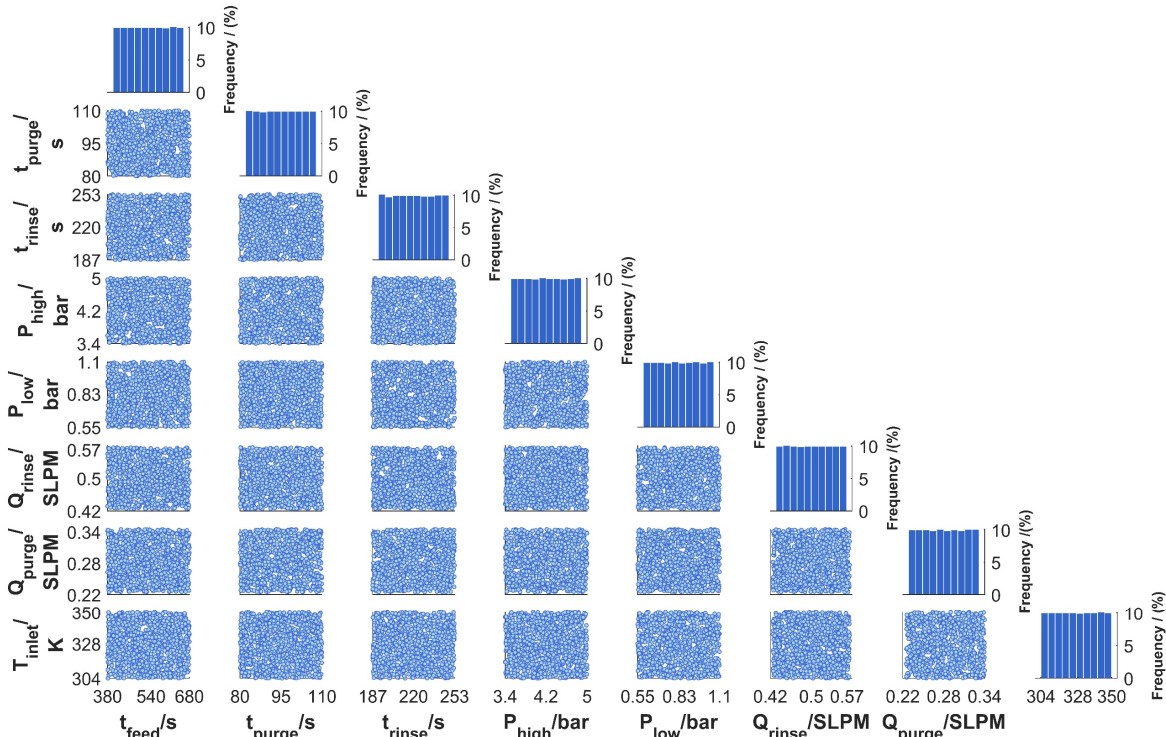

**Figure 4.** Distribution of the inputs samples generated by LHS.

### 3.2. Embedding Dimensions Optimal Selection

He and Asada proposed a methodology [26] to identify the optimal number of past values for the output and input variables in nonlinear dynamic predictors. It consists of calculating the Lipschitz coefficient $q_j^{(m)}$ through Equation (6), using observed input-output data of the system in analysis, to which it is assumed a sensitivity analysis can be done. Through this strategy, it is possible to characterize the relationship of a nonlinear set of input-output relationships for any chaotic/complex dynamic system.

$$q_j^{(m)} = \frac{\left| y_{j-1} - y_j \right|}{\sqrt{(\delta u_1(t-j))^2 + \ldots\ldots + (\delta u_m(t-j))^2}} \tag{6}$$

in which m is the number of observed points in the input-output data, m is the number of input variables to be considered, and $j = 1, 2, \ldots, N$. For each measurement pair, a Lipschitz coefficient is computed, after which each coefficient is then used to calculate the Lipschitz Index $q^{(n)}$ through Equation (7):

$$q^{(n)} = \left( \prod_{k=1}^{p} \sqrt{n} \, q(k)^{(m)} \right)^{(1/p)} \tag{7}$$

in which n is the number of delays considered in the variables, $p$ is a parameter usually between 0.01 N and 0.02 N, and $q(k)^{(n)}$ is the k-th most significant Lipschitz coefficient from all $q_j^{(m)}$ calculated by Equation (6). The method consists of varying n and calculating the correspondent Lipschitz index until its respective value does not differ significantly. Then, the first index to define this region of insensible indexes is the one associated with the desired optimal number of delays.

In this work, this procedure will be used with the NOE and NARX structures to determine the best number of input and output delays to be used in each predictor. The value $p = 0.01$ N was chosen following the literature recommendation [18]. Figure 5

presents the results obtained after applying the Lipschitz analysis on the synthetic data obtained from the virtual plant. It is possible to see that for both cases, one delay for the outputs is enough. On the other hand, four output delays are necessary for the recovery predictor, while five are necessary for the purity predictor.

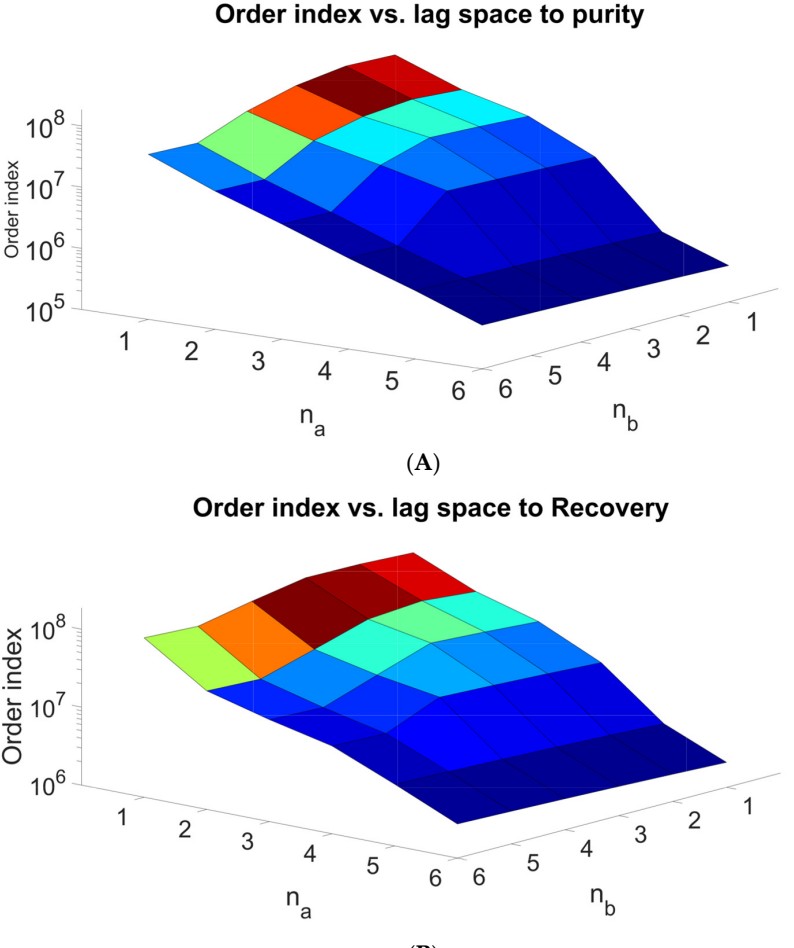

**Figure 5.** Lipschitz indexes for process inputs in order of the recovery (**A**) and purity (**B**), $n_a$ and $n_b$ are the number of past values for the output and input, respectively, as in Equations (3) and (4).

### 3.3. Hyperparameter Tuning

A first step in identifying an artificial neural network model is the definition of the variables that govern the training process and artificial neural network topology. These variables are known as hyperparameters and may have a significant impact on fitting performance. A set of hyperparameters is not directly related to the final model structure, but to how the model will be identified, such as learning rate, momentum, learning rate decay, the number of epochs, and mini-batch size. On the other hand, there is a set of parameters related to the ANN structure, such as the number of layers and neurons, activation function, and layer type. The hyperparameter space is comprised of a set of both discrete and continuous variables, which makes their selection a complex task.

Therefore, an initial optimization step is necessary to select the hyperparameters efficiently. This is performed over the original learning problem, in a way to select the hyperparameters by monitoring the learning cost function [27]. This is a very time-consuming procedure that has been studied in the recent literature. A most recent advance in this field is the HYPERBAND method [28]. This strategy formulates hyperparameter optimization as an exploration problem. A predefined resource is allocated to randomly sampled configu-

rations within a chosen hyperspace. Hence, it requires as input parameters the hyperspace limits and the maximum amount of resources to be used (epochs).

In the present work, the neural networks, training, and hyperparameters tuning were implemented using TensorFlow 2.0 on the Google Colaboratory environment with Python 3.6. Google Colaboratory is a free serverless Jupyter notebook environment for prototyping machine learning models [29]. This allowed performing the identification using hardware accelerators such as tensor process units (TPUs) which can increase neural network training speed in the order of 10 compared to modern CPUs. For the present case, the hyperspace was defined as depicted in Table 1. For the DNN case, two types of deep learning structures are evaluated the gated recurrent unit (GRU) and the long short-term memory (LSTM).

**Table 1.** Hyperparameter search space for RNN, FNN and DNN.

| Hyperparameters | Search Space | | |
|---|---|---|---|
| | **RNN** | **FNN** | **DNN** |
| Initial learning rate | $\{1 \times 10^{-3}, 1 \times 10^{-2}, 1 \times 10^{-1}\}$ | $\{1 \times 10^{-3}, 1 \times 10^{-2}, 1 \times 10^{-1}\}$ | $\{1 \times 10^{-3}, 1 \times 10^{-2}, 1 \times 10^{-1}\}$ |
| Number of recurrent layers | {1, 2, 3} | {1, 2, 3, 4} | {1, 2, 3, 4} |
| Recurrent layer type | - | - | {GRU, LSTM} |
| Number of neurons in the recurrent layers | {100, 150, 250, 300, 350} | 70 to 160, every 10 | {60, 80, 100, 120, 160} |
| Activation function in the recurrent layers | {relu, tanh} | {relu, tanh} | {relu, tanh} |
| Number of neurons in the intermediate fully connected layer | {10, 20, 50, 80, 100} | {70, 90, 100, 120, 130} | {80, 100, 120, 160} |
| Activation function in the fully connected layer | {relu, tanh} | {relu, tanh} | {relu, tanh} |

The results in Table 2 were obtained after applying the HYPERBAND algorithm on the hyperspace presented in Table 1. Interestingly, the algorithm found a mix of GRU and LSTM layers as an optimal structure of DNN for the purity model. On the other hand, only GRU layers are identified as an optimal structure for the recovery model.

**Table 2.** Results of best hyperparameters for each performance indicator for RNN, FNN and DNN.

| Hyperparameters | RNN | | FNN | | DNN | |
|---|---|---|---|---|---|---|
| | $Pur_{CO_2}$ | $Rec_{CO_2}$ | $Pur_{CO_2}$ | $Rec_{CO_2}$ | $Pur_{CO_2}$ | $Rec_{CO_2}$ |
| Initial learning rate | 0.01 | 0.01 | 0.001 | 0.001 | 0.001 | 0.001 |
| Number of recurrent layers | | | - | - | 3 | 2 |
| Recurrent layer type | 1 | 2 | - | - | {Layer 1: GRU, Layer 2: GRU, Layer 3: LSTM} | {Layer 1: GRU, Layer 2: GRU} |
| Number of neurons in the recurrent layers | 250 | {Layer 1: 250, Layer 2: 350} | - | - | {Layer 1: 120, Layer 2: 160, Layer 3: 120} | {Layer 1: 100, Layer 2: 100} |
| Activation function in the recurrent layers | tanh | {Layer 1: relu, Layer 2: relu} | - | - | {Layer 1: relu, Layer 2: relu, Layer 3: tanh} | {Layer 1: tanh, Layer 2: relu} |
| Number of dense in the intermediate fully-connected layer | 1 | 1 | 3 | 4 | 1 | 1 |
| Number of neurons in the intermediate fully-connected layer | 50 | 80 | {Layer 1: 160, Layer 2: 160, Layer 3: 120} | {Layer 1: 150, Layer 2: 150, Layer 3: 150, Layer 4: 150} | 160 | 80 |
| Activation function in the intermediate fully-connected layer | tanh | tanh | {Layer 1: relu, Layer 2: relu, Layer 3: relu} | {Layer 1: relu, Layer 2: relu, Layer 3: relu, Layer 4: relu} | tanh | tanh |

### 3.4. Neural Network Training

Once the hyperparameters are defined, the next step is to train the defined structures of ML models. This is the learning step, where the model's parameters are estimated. The cross-validation method proposed by Schenker and Agarwal [30] was used here to avoid issues related to overtraining and overfitting. This strategy consists in dividing the training data into two separate sets; two groups of networks are trained with each data set. After training is achieved, the data set utilized for training one group of networks is used to validate the other by calculating the mean squared errors. In this way, it is possible to maximize the usage of available information without causing overfitting and overtraining issues.

Further precautions are taken to avoid overtraining. The early stop criterion was used during the training process, that is, the training is stopped after an arbitrary number of iterations or when the mean squared error associated with the training started to increase instead of decrease. The ADAM algorithm was used. Table 3 presents the model's validation performance, where it is possible to observe that the DNN presents the best validation performance. The FNN performance presents a reasonable performance, while the RNN presents a poor performance index compared with the others.

**Table 3.** Models' test performance indexes.

| Metrics | RNN | | FNN | | DNN | |
|---|---|---|---|---|---|---|
| | $Pur_{CO_2}$ | $Rec_{CO_2}$ | $Pur_{CO_2}$ | $Rec_{CO_2}$ | $Pur_{CO_2}$ | $Rec_{CO_2}$ |
| MSE | $7.75 \times 10^{-4}$ | $1.3 \times 10^{-3}$ | $5.104 \times 10^{-5}$ | $1.6223 \times 10^{-4}$ | $1.487 \times 10^{-5}$ | $8.026 \times 10^{-5}$ |
| MAE | 0.0201 | 0.0258 | 0.0039 | 0.0055 | 0.0023 | 0.0032 |

Figures 6–8 portray the prediction of the FNNs, RNNs, and DNNs against the virtual plant. Thus, it is possible to visualize the models' behavior during the test step. The behavior observed in these graphics reflects the performance index obtained during the ML models identification. Hence, it is possible to see a good adherence of the DNN to the test data, a fair adherence by the FNN, and a poor prediction by the RNN.

Finally, a better analysis of the validation results can be performed based on the parity graphics shown in Figure 9. The parity plots display the ANNs' predictions versus the observed values in the synthetic data (Figure 9). For the cases of the DNN and the FNN models, it is possible to observe that their parity is randomly distributed around the diagonal line across the whole range of values, thus indicating that the residuals are random. This demonstrates that the model was satisfactorily estimated within that domain. On the other hand, for the RNN models, no randomness is observed in the parity graphics.

This, therefore, evidences the problems of the solely recurrent approach. These are well-known issues in this field, as mentioned in the Introduction. These issues are the reason for the widespread use of FNNs-based on NARX predictors. Furthermore, the good performance of the FNN-NARX approach led to the reluctant employment of DNN strategies in the chemical processes literature [18,21]. On the other hand, these models are usually evaluated only by their test behavior. After that, the model is considered a good tool to be employed in the practical case. However, if they are employed to address a situation where they are not adequately used, their quality will quickly degenerate. This issue will be further discussed in the results of this work. Overall, this is a practical observation of the topics discussed here.

It is important to note that there are significant differences between the computation time to train an NOE model and a NARX model. In the present work, it took approximately 1.8 s to perform a single training epoch for the NARX model. On the other hand, it took 3.3 s to perform a training epoch for the NOE model. Overall, the full training of the NARX took 278 s, while the full training of the NOE model took 506 s. The trainings were done in an online cloud computing service. However, the training is a step performed offline, and

it should not be a critical issue. Still, this point should be considered when choosing the most suitable strategy.

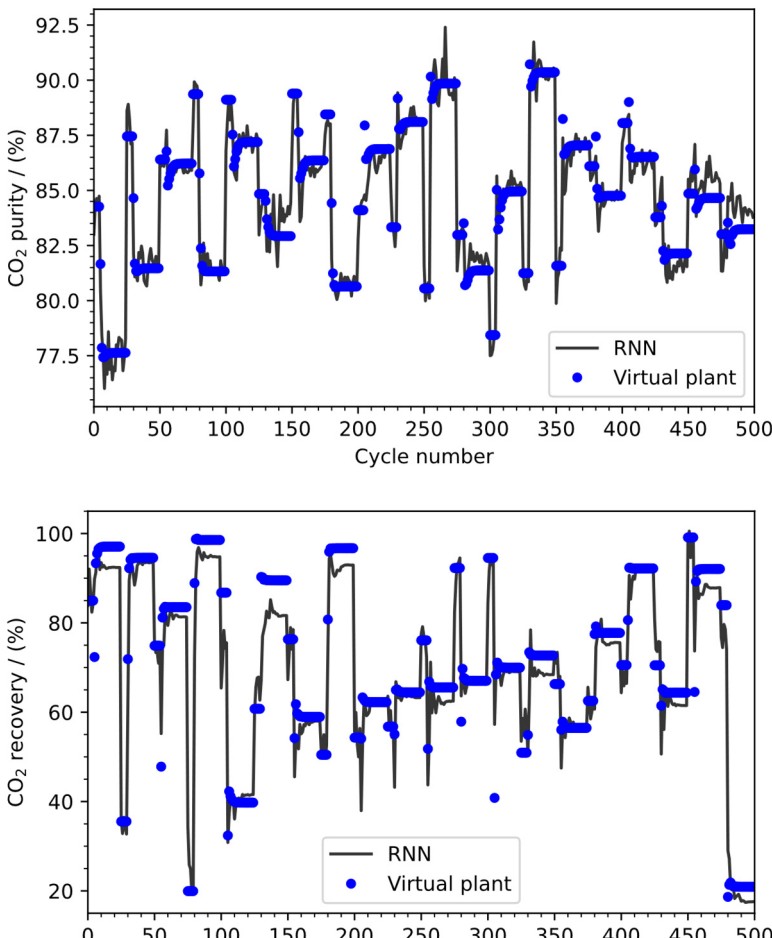

**Figure 6.** RNN test performance.

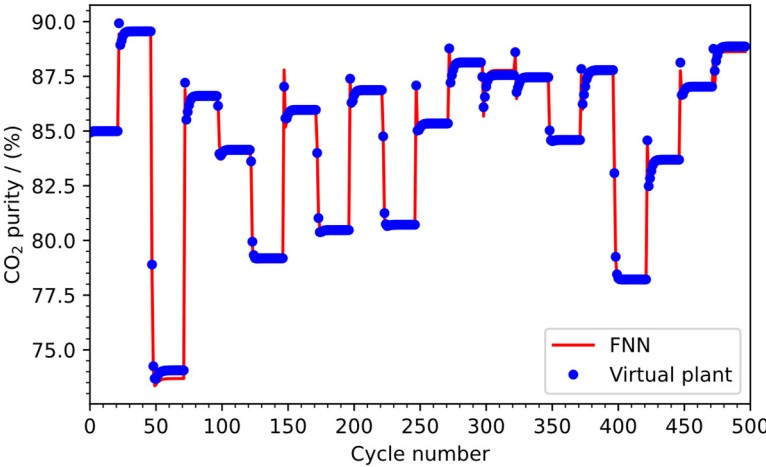

**Figure 7.** *Cont.*

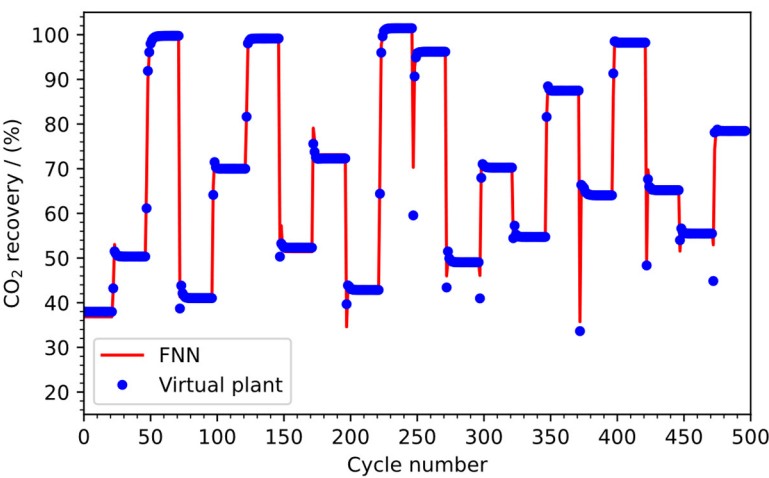

**Figure 7.** FNN test performance.

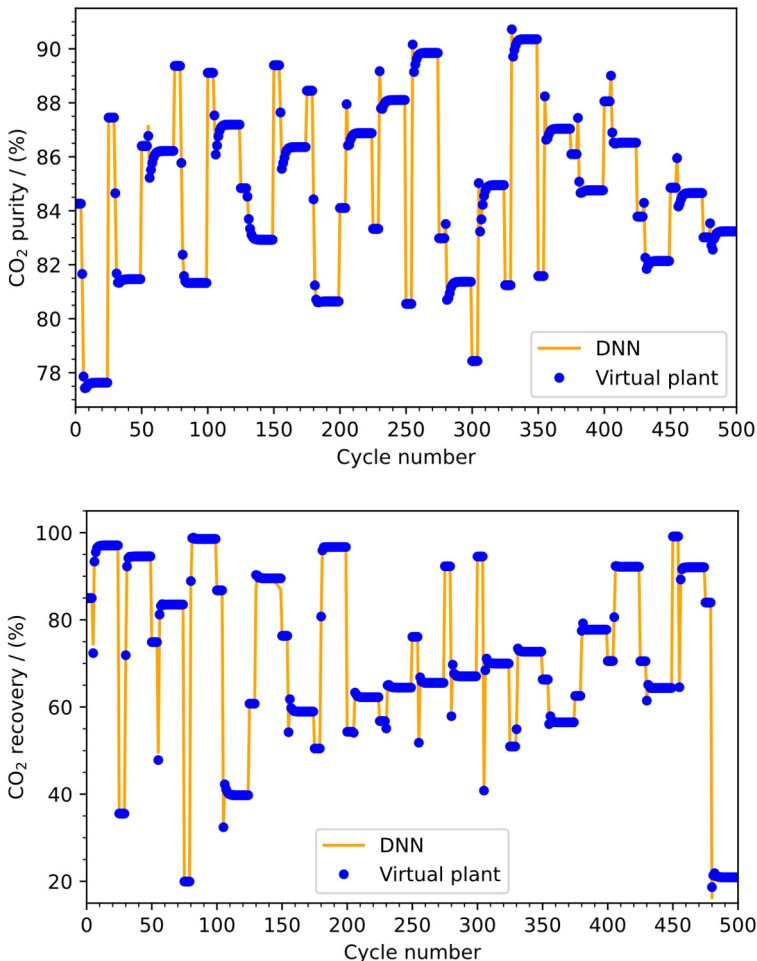

**Figure 8.** DNN test performance.

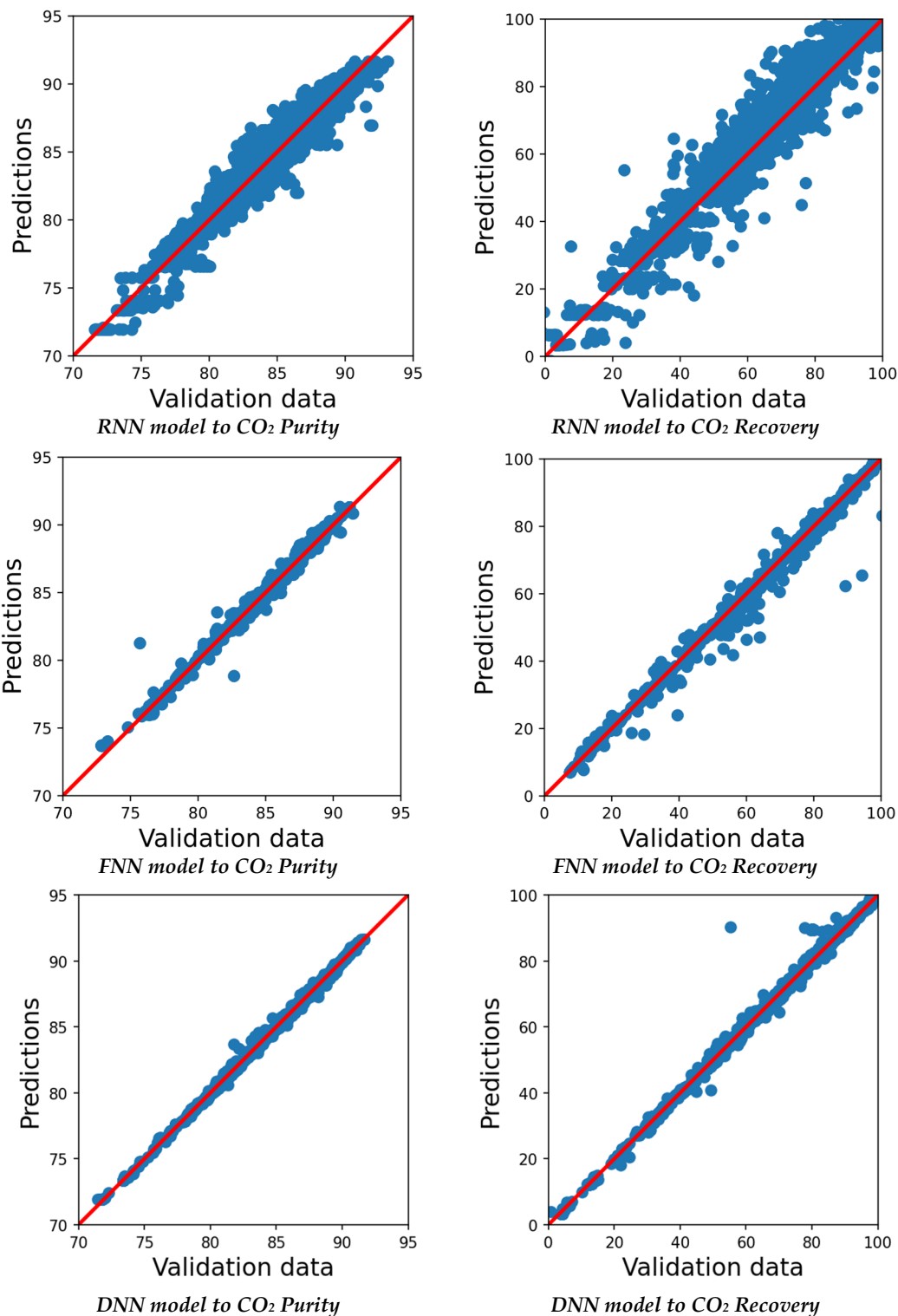

**Figure 9.** Models' validation parity graphics.

### 3.5. Prediction Case

In this case, a virtual analyzer to provide real-time PSA purity and recovery predictions is proposed. Following the problem discussed here, three analyzers are developed: a NARX based FNN soft-sensor, and two other NOE-based (RNN and DNN) soft sensors.

The concentration measurement in PSA usually presents a significant deadtime, which can reach several minutes. Hence, the purity and recovery are unknown while no mea-

surement is available. Therefore, the virtual analyzer for these units is a suitable solution, providing information regarding the process variables while no measurement is available. As the NARX structure presents the exogenous input, this becomes a gate to introduce the purity and recovery measured values when the measurement is available. Hence, it is expected that the FNN-NARX sensor will produce the best performance in this case. Therefore, the NARX is used to perform a short-term prediction while new information about the system is collected. On the other hand, the RNN and DNN are misemployed here, as we are dealing with a short-prediction scenario. The results compare the measured values provided by the phenomenology model with the values predicted by the structures. Figure 10 presents the results obtained for each structure to perform the predictions of purity and recovery. The simulated measurement deadtime was 10 cycles. As can be seen, the NOE, whose structures are more adequate to simulations, provides a more significant difference between the predicted values and the reference values when compared with the NARX predictor. This shows that the NARX, in this short-term prediction case, is indeed adequate to provide information of the system: the error cumulation is not detrimental in the prediction horizon chosen.

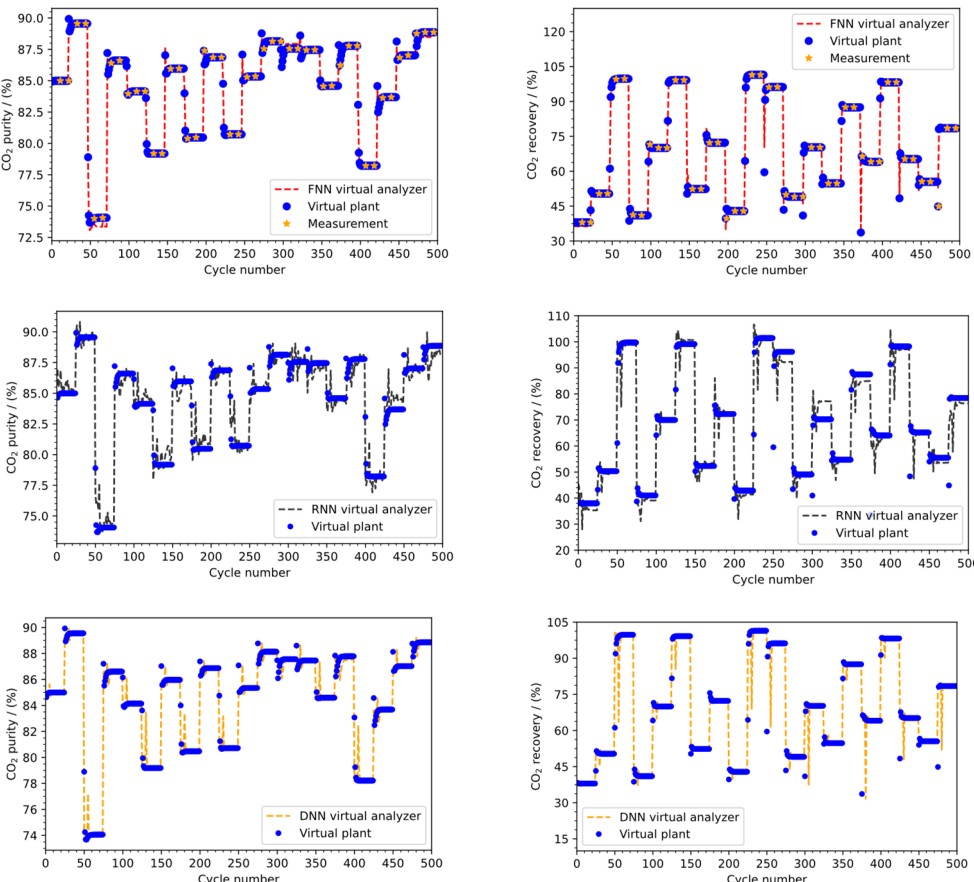

**Figure 10.** Soft sensors evaluation for real-time PSA recovery and purity prediction.

Further comparison is provided by Table 4, which presents the performance indexes of each soft sensor developed. The results presented in this table corroborate the discussions made above.

Finally, these results provide the PSA soft sensor, which is one contribution of this work. As it is possible to see from the results, the soft sensor can efficiently provide information regarding the system behavior while there are no measurements. Furthermore, as it is possible to see from the virtual plant, the soft sensor can actually provide precise real-time information.

**Table 4.** Soft sensors performance indicators.

|  | MAE | | MSE | |
|---|---|---|---|---|
|  | **Purity** | **Recovery** | **Purity** | **Recovery** |
| RNN | 0.6352 | 2.9592 | 0.9306 | 29.8229 |
| DNN | 0.2919 | 1.5503 | 0.4940 | 31.2703 |
| FNN | 0.1854 | 0.8084 | 0.0572 | 1.5549 |

*3.6. Simulation Case*

Process simulation within a long horizon is usually necessary for process control and optimization. For example, in dynamic optimization where it is necessary to evaluate the evolution of the process from a given steady-state to another or in the infinity prediction horizon in model predictive controllers. As discussed here, the NARX structures are not recommended in these situations. However, it is usual to see them being employed for these cases in the literature. Therefore, as another case study, the models identified here were used to perform simulations of the PSA dynamic evolution. Figure 11 portrays the result of the simulations. A total of five different operating conditions were given to the PSA plant, and sufficient time was given until the system reached the corresponding cyclic steady-state.

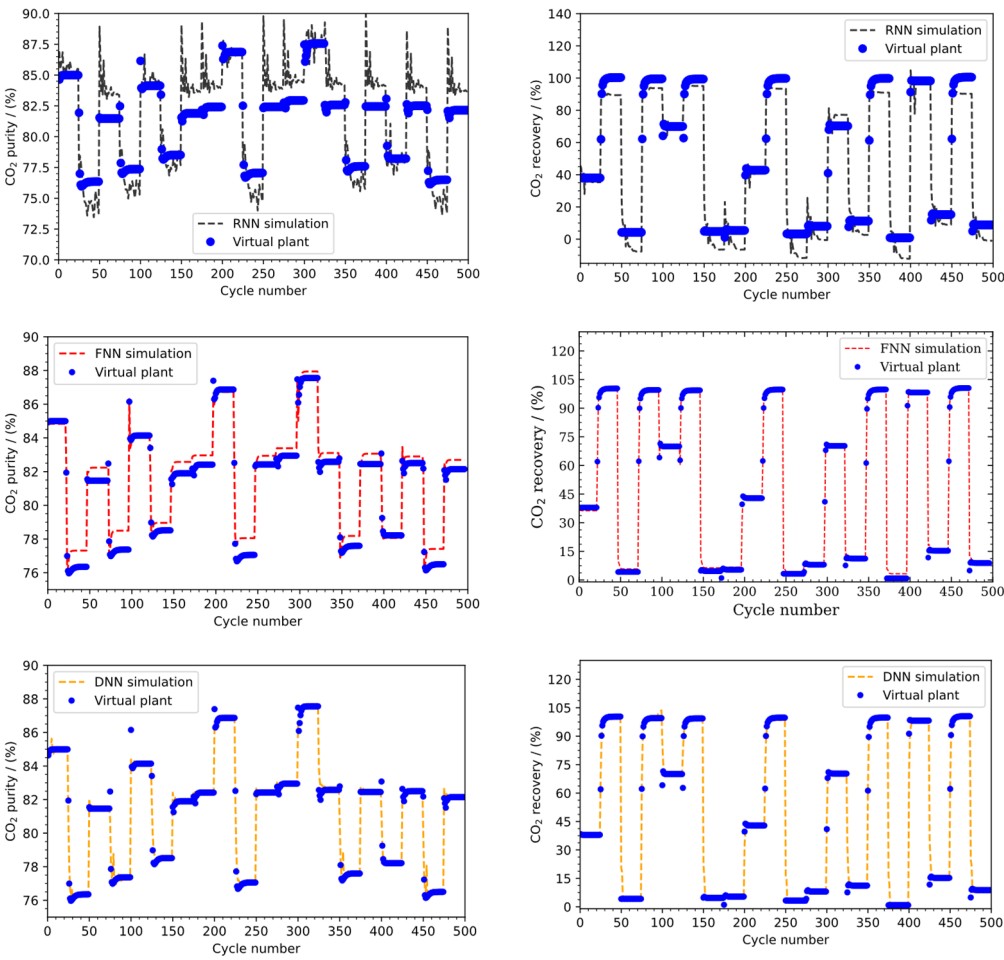

**Figure 11.** Simulation of the PSA operating from several different operating conditions.

As it is possible to see from the graphics in Figure 11, the FNN-NARX model presented an offset at some conditions. On the other hand, the DNN model efficiently predicted the cyclic steady-state and the system dynamics. A series of 100 operating conditions were

given to the model to better evaluate the differences between the ML strategies under a simulation scenario. The graphics are not provided for this last case, as the visualization is not appropriated due to the data density. Table 5 provides the simulation performance indexes. It is possible to see that the DNN is significantly superior to the FNN in the simulations scenario. The performance indexes differences are a clear demonstration of how the FNN-NARX performance can degenerate along a simulation.

**Table 5.** Simulation performance indicators.

| | MAE | | MSE | |
|---|---|---|---|---|
| | **Purity** | **Recovery** | **Purity** | **Recovery** |
| RNN | 1.7159 | 108.0981 | 5.1987 | 8.3898 |
| DNN | 0.1799 | 0.8688 | 0.0740 | 4.8250 |
| FNN | 0.7135 | 2.1710 | 0.6541 | 10.2519 |

## 4. Conclusions

This work addresses the problem of identifying machine learning models for simulations and prediction purposes. It presents a comprehensive guideline for the ML models identification. On the other hand, it also deals with the online measurement issue found in a pressure swing adsorption unit. Thus, a series of ML models are identified to model the PSA process.

It is important to note that the models employed in the simulation and the prediction scenario are the same; their application is different. It is worth mentioning that the DNN presented the best performance for the test of the ML models during the model identification. Therefore, it was demonstrated how misleading it is to use the test performance solely without considering their application. The model with the best test performance (DNN) had inferior behavior during the prediction. On the other hand, the model with acceptable performance in the test (FNN-NARX) had poor behavior when applied for simulation purposes. Therefore, it was demonstrated the proper evaluation of the kind of application a machine learning model is sought for is critical.

Regarding the online measurement, the results demonstrated that the soft sensor based on FNN-NARX can efficiently predict the states while there is no measured data. On the other hand, a model for simulation purposes based on DNN was provided, showing to be able to efficiently simulate the model dynamic behavior.

**Author Contributions:** Conceptualization, I.B.R.N., C.M.R., P.H.M.; methodology, I.B.R.N., C.M.R., P.H.M., V.V.S., E.A.C.; writing—original draft preparation, I.B.R.N., C.M.R., E.A.C., V.V.S.; writing—review and editing, I.B.R.N., C.M.R., and A.M.R.; supervision, I.B.R.N., A.E.R., A.M.R. All authors have read and agreed to the published version of the manuscript.

**Funding:** This work was financially supported by: Project—NORTE-01-0145-FEDER-029384 funded by FEDER funds through NORTE 2020—Programa Operacional Regional do NORTE—and by national funds (PIDDAC) through FCT/MCTES. This work was also financially supported by: Base Funding—UIDB/50020/2020 of the Associate Laboratory LSRE-LCM—funded by national funds through FCT/MCTES (PIDDAC), Capes for its financial support, financial code 001 and FCT—Fundação para a Ciência e Tecnologia under CEEC Institucional program.

**Conflicts of Interest:** The authors declare no conflict of interest.

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
