# Peer review of "Machine Learning-Based Dynamic Modeling for Process Engineering Applications: A Guideline for Simulation and Prediction from Perceptron to Deep Learning"

_processes, doi:10.3390/pr10020250_

Round 1
Reviewer 1 Report
Submission possesses solid scientific ground for publication. However, there are some serious drawbacks that should be taken into account:
- Introduction is too general and not providing any literature review. The purpose of the literature review is to provide knowledge gap and form the hypotheses for your research.
- There is relatively small amount of results compared to the size of methods section.
- References seem outdated, because most of the references are older than 5 years. Please make state of the art literature review covering recent research within this topic
- Only one reference to this journal means that authors aren't aware of knowledge base already present in this journal or that the work is out of journal scope.
- Multiple "Error! Reference source not found" in the text
- Abbreviations should be defined upon first appearance in the text, i.e. MAE and MSE
- Lines 245-247: multiple repetition of the same information in the manuscript body is not necessary
- Each reference should be explained by at least one separate sentence and not pilled up like in line 132, 135 etc.
- Description under Figure 1 is too lengthy
- Avoid consecutive section headings without anything in between. Either merge your sections or provide short paragraph of introductory text.
Author Response
Thank you. Please see attached file.

Reviewer 2 Report
This paper studies the application of ML for modeling dynamic systems and compares the differences in performance depending on the model structure (whether is a NARX model structure or a NOE model structure) and the modelling purpose (whether the objective is prediction or simulation). Different neural networks classes are implemented in an interesting case study corresponding a PSA process.
Here some comments:
- In my opinion, a brief introduction to PSA should be given in section 2.1. For any reader which is not familiar with PSA, it might be difficult to understand the operation of the unit.
- I would recommend a native English speaker to proofread the paper, e.g., “structuration” (as in “data structuration”) does not seem to be proper English.
- It seems to be a problem while generating the pdf file. References to equations are lost. This makes difficult to read the paper and to link the text content with the equations. The paper cannot be published without solving this problem.
- The notation in equations (5) and (6) is not completely consistent. Section 2.8 must be better explained.
- Table 2 is misleading as the column names are not consistent with the column names in Table 1.
Author Response
Thank you. Please see attached file.

Reviewer 3 Report
This paper presented machine learning-based dynamic modeling for process engineering applications. This work mainly showed a virtual analyzer for a Pressure Swing Adsorption (PSA) unit. However, for the current version of the paper, I do not think that this paper can be published in Processes.
1 - Please double check your manuscript title. "a novel Virtual Analyze"?
2 - In Abstract, what are the main novelty in this work compared with previous works? As this work was focused on PSA, more contents on the ML applications on PSA should be studied and discussed.
3 - In Introduction section, there are only 8 references cited. Please demonstrate the latest developments in using ML on PSA processes. There are a lot of papers published in this field such as Machine learning-based multiobjective optimization of pressure swing adsorption, Machine learning–based optimization for hydrogen purification performance of layered bed pressure swing adsorption, Machine learning model and optimization of a PSA unit for methane-nitrogen separation, etc. What are your major advantages compared with these publications? With a better prediction? The ML methods have been widely reported in the literature.
4 - Line 111, "H2/CO stoichiometric ratio between 2.2 and 2.3" -- any reference for the values?
5 - In Line 115, what are the improvements in your model compared with the phenomenological model reported by Silva et al. in 1999?
6 - There are several citations showing "Error! Reference source not found".
7 - Please provide more details of the training data used in this work.
8 - In Figure 5, please give the legend for the surface maps.
9 - All this work is based on the reported model from the literature. How could you improve the prediction accuracy? Any limitations for the applications of the model? How could you use the model to predict the future CO2 purity and recovery results? For different plants? How could your work to help identifying the optimal conditions?
10 - In Table 5, DNN showed the best performance. However, this is only for this case?
Author Response
Thank you. Please see attached file.

Round 2
Reviewer 1 Report
Authors should make language check, correct some minor mistakes such as multiple spaces between words and correctly reference Figures and Equations. Currently, there are multiple: "Error! Reference source not found.". Please convert your text to pdf before uploading.
Authors should also include comparison of calculation time between NARX and NOE approach, giving the fact that NOE comes with the price of highly increased computational expenses compared to NARX. Is it worth it?
Author Response
We want to thank the reviewer for the support and contributions made to improve our work. Please see attached file.

Reviewer 2 Report
It is nice to see that the recommendations were applied.
Author Response
We want to thank the reviewer for the support and contributions made to improve our work.
Reviewer 3 Report
The authors have revised the manuscript according to the previous comments. It is good to accept now.
Author Response

(The authors gave the same response as above.)
